# Parameter Estimation Algorithm of Frequency-Hopping Signal in Compressed Domain Based on Improved Atomic Dictionary

**DOI:** 10.3390/s23115065

**Published:** 2023-05-25

**Authors:** Weipeng Zhu, Yourui Wang, Hu Jin, Yingke Lei

**Affiliations:** Electronic Countermeasure Institute, National University of Defense Technology, Hefei 230037, China; m13357188541@163.com (W.Z.); wyr_nudt@nudt.edu.cn (Y.W.); zhangzhi@nudt.edu.cn (H.J.)

**Keywords:** segment, compressive sampling, maximum dot product, improved atomic dictionary, parameter estimation

## Abstract

This paper considers the problem of estimating the parameters of a frequency-hopping signal under non-cooperative conditions. To make the estimation of different parameters independently of each other, a compressed domain frequency-hopping signal parameter estimation algorithm based on the improved atomic dictionary is proposed. By segmenting and compressive sampling the received signal, the center frequency of each signal segment is estimated using the maximum dot product. The signal segments are processed with central frequency variation using the improved atomic dictionary to accurately estimate the hopping time. We highlight that one superiority of the proposed algorithm is that high-resolution center frequency estimation can be directly obtained without reconstructing the frequency-hopping signal. Additionally, another superiority of the proposed algorithm is that hopping time estimation has nothing to do with center frequency estimation. Numerical results show that the proposed algorithm can achieve superior performance compared with the competing method.

## 1. Introduction

With the continuous upgrading of communication countermeasures, there have been increasing demands for reconnaissance of frequency-hopping (FH) signals from non-cooperative parties. Obtaining the parameters of non-cooperative FH signals is a crucial link in communication countermeasures; as a result, FH signal parameter estimation has become a hot topic in the realm of communication countermeasures.

At present, research on FH signal parameter estimation is mainly based on three methods: time–frequency analysis, atomic decomposition and compressed sensing. The methods of time–frequency analysis mainly include linear transformations such as short-time Fourier transform (STFT), Gabor transform (GT) and wavelet transform (WT), as well as quadratic time–frequency distributions such as Wigner–Ville distribution (WVD), Pseudo Wigner–Ville distribution (PWVD) and smooth Pseudo Wigner–Ville distribution (SPWVD). Due to the fixed window function, STFT is unable to take time resolution and frequency resolution into consideration at the same time [1]. GT can be considered as the optimal STFT, but it is unable to adjust the type and length of window function according to frequency [2]. WT can present different time–frequency resolutions at different frequencies, but it has poor performance on FH signals [3]. The time–frequency resolution of WVD can reach the peak value in theory, but its application to multicomponent signals is limited because of the cross term [4]. PWVD can reduce the impact of the cross term with the decline in time–frequency resolution [5]. SPWVD can completely eliminate the impact of the cross term, but it reduces the time–frequency resolution compared to PWVD [6]. In addition, in order to improve the accuracy of parameter estimation for FH signals under underdetermined mixing, a combined time–frequency analysis method combining STFT and SPWVD is proposed [7]. Compared with existing time–frequency methods based on time–frequency sparsity, the parameter estimation error is significantly reduced with a small increase in complexity.

Based on the atomic dictionary, atomic decomposition decomposes a signal into a combination of several atoms. In a study by Fan et al., the atomic dictionary with three parameters was established based on this principle to achieve high accuracy estimation under low signal-to-noise ratio (SNR) conditions [8]. However, the dimension of this atomic dictionary is so large that the computational complexity is quite high. In order to eliminate the high redundancy of atomic decomposition effectively, the authors in [9] modified the three-parameter atomic dictionary into the single-parameter atomic dictionary. However, the frequency of atom matching causes its high computational complexity.

The methods of time–frequency analysis and atomic decomposition are based on Nyquist’s theorem, which causes a large amount of sampling data and high computational complexity. To reduce the amount of sampling data, compressed sensing is proposed. Sparse reconstruction is a traditional processing way in the theory of compressed sensing [10], the algorithm of reconstruction not only increases the amount of computation but is also sensitive to noise. To overcome the shortcomings of sparse reconstruction, some scholars have proposed algorithms for estimating FH signal parameters directly in the compressed domain using the sparse characteristics of FH signal [11,12,13,14,15,16,17,18,19,20,21,22]. Zhang et al. established a redundant atomic dictionary to estimate hopping time accurately [23], which reduces the computational complexity compared to the study in [9]. However, the redundant atomic dictionary is based on center frequency estimation, which makes the accuracy of hopping time estimation dependent on the accuracy of center frequency estimation.

To reduce the amount of sampling data and correlation between hopping time estimation and center frequency estimation, a parameter estimation algorithm of FH signal in compressed domain based on improved atomic dictionary is proposed in this study.

The main contributions of this study are as follows:(1)There is no need to reconstruct the FH signal; frequency can be estimated in the compressed domain.(2)An improved atomic dictionary is constructed to estimate hopping time accurately. The dictionary has a simple structure and small dimensions, and the dictionary is not based on frequency estimation, making the accuracy of hopping time estimation independent of the accuracy of frequency estimation.(3)Mathematical proof is given for the feasibility of accurately estimating the hopping time using the improved atomic dictionary.

The remainder of the paper is organized as follows. FH signal model and compressed sampling model are built in Section 2, and the parameter estimation algorithm based on improved atomic dictionary is proposed in Section 3. Section 4 presents tests and data analysis, followed by the mechanism elucidation. Finally, the study is concluded in Section 5.

## 2. Problem Description and Models

### 2.1. FH Signal

An FH signal is a non-stationary signal whose frequency changes over time, and its carrier frequency is controlled by a pseudorandom sequence. The mathematical model of FH signal received by the communication receiver can be expressed as follows:(1)$$x(t)=\sqrt{2P}\sum_{i=0}^{N_{h}}rect_{T_{h}}(t-iT_{h}-t_{0})e^{j2\pi f_{i}(t-iT_{h}-t_{0})+j\theta_{i}}m(t)+n(t)+I(t)$$
where p is the signal power, $N_{h}$ is the frequency of hopping, $T_{h}$ is the period, $rect_{T_{h}}(t)$ is the rectangular window with width $T_{h}$, $t_{0}$ is the original time of hopping, $f_{i}$ is the center frequency at hop *i*, $\theta_{i}$ is the phase at hop *i*, m(t) denotes an information modulation term multiplied to the FH signal, n(t) is the noise, and I(t) is the interference.

### 2.2. Compressed Sampling

If a signal has only a small number of non-zero values in a certain domain, it is sparse in that domain, and it can be reconstructed from a small number of sampling points that are far below the requirements of Nyquist’s theorem. The compressed sampling model is as follows:(2)y=Φx
where *x* is the *N*-dimensional digital signal, *y* is the *M*-dimensional digital signal, and Φ is the measurement matrix, M≪N.

If *x* signal has sparsity in a certain transformation domain, the sparse representation of the *x* signal is given by:(3)x=Ψs
where Ψ is the sparse base matrix, and *s* is the sparse coefficient vector.

Substituting (3) into (2), we can rewrite *y* as: (4)y=ΦΨs=Θs
where Θ is the sensing matrix.

## 3. Algorithm Description

The algorithm in this paper is divided into three parts: data preprocessing, center frequency estimation and hopping time estimation. In the process of data preprocessing, the FH signal is segmented, and compressed sampling is applied to each signal segment. During frequency estimation, the method of maximum dot product is used to estimate the center frequency of each signal segment. During the hopping time estimation, after performing differential processing on the center frequency of each signal segment, an improved atomic dictionary is used to estimate hopping time.

### 3.1. Data Preprocessing

In this process, FH signal is segmented, and each segment contains only a portion of a period. If the signal segment does not contain the hopping point, it has only one frequency component. On the contrary, the signal segment has two frequency components when it contains the hopping point. 

After that, compressed sampling is applied to each segment. Sparse decomposition of each segment according to Equation (4) is given by:(5)$$y_{p}=\Theta_{p}s_{p}$$
where $y_{p}$ is the signal segment after compressed sampling, $\Theta_{p}$ is the sensing matrix that consists of a Gaussian random measurement matrix and Fourier sparse base matrix, and $s_{p}$ is the sparse coefficient vector of the signal segment.

### 3.2. Center Frequency Estimation Based on Maximum Dot Product

After preprocessing, the method of maximum dot product is used to estimate the center frequency. The position of the largest non-zero coefficient in the sparse coefficient vector can be expressed as follows:(6)$$pos=argmax\left(\left|\langle \Theta_{p},y_{p}\rangle \right|\right)$$

According to the theory of compressed sensing, the relation between position and center frequency can be expressed as:(7)$$f=\frac{(pos-1)f_{s}}{N_{p}}$$
where $f_{s}$ denotes the sampling rate, and $N_{p}$ denotes the original length of the signal segment.

For the signal segment without one hopping point, the estimated value of frequency is frequency of the signal segment, and for the signal segment with one hopping point, the estimated value of frequency is the frequency of the signal segment before or after hopping.

To ensure the accuracy of frequency estimation, the length of each signal segment should be as large as possible within the allowable range. The larger the length of each signal segment, the lower the time resolution of the hopping time. To improve the time resolution of the hopping time, an improved atomic dictionary is proposed.

### 3.3. Hopping Time Estimation Based on Improved Atomic Dictionary

After frequency estimation, differential processing is performed on the center frequency of each signal segment to detect the signal segment that has two frequency components. The hopping point exists in this signal segment and the previous signal segment, so these two signal segments are selected as one observation object.

If the Fourier sparse base matrix is used directly, it can only reflect the characteristics of the FH signal in the frequency domain. To fully reflect the characteristics of the FH signal in the time–frequency domain, an improved atomic dictionary is constructed based on Fourier sparse base matrix, as shown in Figure 1a. The purpose of the improved atomic dictionary is to analyze the duration of each hop signal and construct a Fourier sparse base matrix corresponding to each hop signal, and then arrange these Fourier sparse base matrices on the main diagonal chronologically to fully reflect the time–frequency characteristics. The signal segment containing one hopping point has only two hop signals, and the improved atomic dictionary at this condition is shown in Figure 1b.

The improved atomic dictionary at this condition can be expressed as follows:(8)$$\Psi_{p}=\left(\begin{array}{cc} \Psi_{K\times K} & 0_{K\times (N-K)}\\ 0_{(N-K)\times K} & \Psi_{\left(N-K)\times (N-K\right)} \end{array}\right)$$

As is shown in Equation (8), these two Fourier sparse base matrices $\Psi_{K\times K}$ and $\Psi_{\left(N-K)\times (N-K\right)}$ will change as K changes, K=1,2,⋯,N−1. Sparse decomposition of observed object $x_{p}$ on the improved atomic dictionary $\Psi_{p}$ is for obtaining the sparse coefficient vector α:(9)$$x_{p}=\Psi_{p}\alpha$$

There are two large non-zero coefficients $\alpha_{1}$ and $\alpha_{2}$ on the sparse coefficient vector α. As K changes, ${\left|\alpha_{1}\right|}^{2}+{\left|\alpha_{2}\right|}^{2}$ also changes.

**Theorem** **1.***When* K *arrives at the position of the hopping point,* ${\left|\alpha_{1}\right|}^{2}+{\left|\alpha_{2}\right|}^{2}$ *will take the maximum value*.

**Proof** **of** **Theorem** **1.**Assume that the observation object contains the signal segment $x_{N_{1}}$ with $N_{1}$-length, $w_{1}$-frequency, $\theta_{1}$-phase and the signal segment $x_{N_{2}}$ with $N_{2}$-length, $w_{2}$-frequency, and $\theta_{2}$-phase. After sparse decomposition through improved atomic dictionary $\Psi_{p}$, $\alpha_{1}$ and $\alpha_{2}$ can be expressed as follows:
(10)$$\left\{\begin{array}{l} \alpha_{1}=\frac{e^{j\theta_{1}}}{2\pi }K_{1}\times Sa\left(\frac{\left(w-w_{1}\right)K_{1}}{2}\right),0<K_{1}<N_{1}\\ \alpha_{2}=\frac{e^{j\theta_{2}}}{2\pi }K_{2}\times Sa\left(\frac{\left(w-w_{2}\right)K_{2}}{2}\right),0<K_{2}<N_{2} \end{array}\right.$$
where *Sa*(*x*) *= sin*(*x*)*/x*.

While $0<K<N_{1}$, $\alpha_{1}$ and $\alpha_{2}$ are as follows:(11)$$\left\{\begin{array}{l} \left|\alpha_{1}\right|=\left|\frac{e^{j\theta_{1}}}{2\pi }K\times Sa\left(\frac{\left(w-w_{1}\right)K}{2}\right)\right|=\frac{K}{2\pi },w=w_{1}\\ \left|\alpha_{2}\right|<\left|\frac{e^{j\theta_{2}}}{2\pi }N_{2}\times Sa\left(\frac{\left(w-w_{2}\right)N_{2}}{2}\right)\right|=\frac{N_{2}}{2\pi },w=w_{2} \end{array}\right.$$
At this moment, ${\left|\alpha_{1}\right|}^{2}+{\left|\alpha_{2}\right|}^{2}$ is as follows:(12)$${\left|\alpha_{1}\right|}^{2}+{\left|\alpha_{2}\right|}^{2}<\frac{K^{2}+N_{2}^{2}}{4\pi^{2}}<\frac{N_{1}^{2}+N_{2}^{2}}{4\pi^{2}}$$
While $K=N_{1}$, $\alpha_{1}$ and $\alpha_{2}$ are as follows:(13)$$\left\{\begin{array}{l} \left|\alpha_{1}\right|=\left|\frac{e^{j\theta_{1}}}{2\pi }N_{1}\times Sa\left(\frac{\left(w-w_{1}\right)N_{1}}{2}\right)\right|=\frac{N_{1}}{2\pi },w=w_{1}\\ \left|\alpha_{2}\right|=\left|\frac{e^{j\theta_{2}}}{2\pi }N_{2}\times Sa\left(\frac{\left(w-w_{2}\right)N_{2}}{2}\right)\right|=\frac{N_{2}}{2\pi },w=w_{2} \end{array}\right.$$
At this moment, ${\left|\alpha_{1}\right|}^{2}+{\left|\alpha_{2}\right|}^{2}$ is as follows:(14)$${\left|\alpha_{1}\right|}^{2}+{\left|\alpha_{2}\right|}^{2}=\frac{N_{1}^{2}+N_{2}^{2}}{4\pi^{2}}$$
While $N_{1}<K<N_{1}+N_{2}$, $\alpha_{1}$ and $\alpha_{2}$ are as follows:(15)$$\left\{\begin{array}{l} \left|\alpha_{1}\right|<\left|\frac{e^{j\theta_{1}}}{2\pi }N_{1}\times Sa\left(\frac{\left(w-w_{1}\right)N_{1}}{2}\right)\right|=\frac{N_{1}}{2\pi },w=w_{1}\\ \left|\alpha_{2}\right|=\left|\frac{e^{j\theta_{2}}}{2\pi }\left(N_{1}+N_{2}-K\right)\times Sa\left(\frac{\left(w-w_{2}\right)\left(N_{1}+N_{2}-K\right)}{2}\right)\right|=\frac{\left(N_{1}+N_{2}-K\right)}{2\pi },w=w_{2} \end{array}\right.$$
At this moment, ${\left|\alpha_{1}\right|}^{2}+{\left|\alpha_{2}\right|}^{2}$ is as follows:(16)$${\left|\alpha_{1}\right|}^{2}+{\left|\alpha_{2}\right|}^{2}<\frac{N_{1}^{2}+{\left(N_{1}+N_{2}-K\right)}^{2}}{4\pi^{2}}<\frac{N_{1}^{2}+N_{2}^{2}}{4\pi^{2}}$$
To sum up, while $K=N_{1}$, ${\left|\alpha_{1}\right|}^{2}+{\left|\alpha_{2}\right|}^{2}$ will take the maximum value $\frac{N_{1}^{2}+N_{2}^{2}}{4\pi^{2}}$. □

We obtain the sparse coefficient vector α according to the following equation:(17)$$\left\{\begin{array}{l} y_{p}=\Phi x_{p}=\Phi \Psi_{p}\alpha =\Theta_{p}\alpha \\ \alpha =OMP(y_{p},\Theta_{p},2) \end{array}\right.$$
where *OMP* denotes the algorithm of orthogonal matching pursuit [24].

### 3.4. Computational Complexity Analysis

The computational complexity of an algorithm mainly depends on the number of floating-point multiplication operations, so this article uses the number of floating-point multiplication operations to characterize the computational complexity.

For frequency estimation, the original length of each segment is $N_{p}$, compression ratio is k, and the length after compression of each segment is given by $N_{p}/k$. Therefore, the number of floating-point multiplication operations required to obtain the position corresponding to the maximum dot product of each segment is $N_{p}^{2}/k$. Considering that the number of segments is L, the computational complexity of frequency estimation can be denoted as $N_{p}^{2}L/k$.

For hopping time estimation, denote the original length of each observation object as $2N_{p}$ and the length after compression of each observation object as $2N_{p}/k$, so the number of floating-point multiplication operations required is $4N_{p}^{2}/k$. According to Equation (17), obtaining the sparse coefficient vector corresponding to each point needs to match twice, and there are $M_{p}$ points in each observation object; therefore, the number of floating-point multiplication operations required to estimate accurately the hopping time of each observation object is $8M_{p}N_{p}^{2}/k$. Considering the number of observation object is B, the computational complexity of hopping time estimation can be denoted as $8BM_{p}N_{p}^{2}/k$.

In conclusion, the computational complexity of the proposed algorithm can be expressed as $\left(L+8BM_{p}\right)N_{p}^{2}/k$.

## 4. Experiments

Experiments are carried out to verify the effectiveness of the proposed algorithm, which are divided into five groups. The first group will verify the principle of hopping time estimation to lay the foundation for subsequent experiments. The second group will analyze the impact of additive white Gaussian noise (AWGN) on the performance of proposed algorithm and provide a mechanism explanation for the impact of AWGN. There are many interferences in the actual channel environment, so the third group will simulate and analyze the performance changes in the algorithm under the interference of fixed-frequency signals. The search step size and compression ratio are two key parameters in this algorithm, so the fourth group will analyze the impact of these two parameters on the performance of the proposed algorithm. Finally, the fifth group will compare the proposed algorithm with previous work to indicate that the proposed algorithm improves estimation performance while reducing computational complexity.

### 4.1. Principle Verification

Table 1 lists the FH signal parameters that are used in the simulation. 

At the stage of frequency estimation, set the original length of each segment as 2500 points and the compression rate as 5, so the length after compression of each segment is set to 500 points. During the process of compressed sampling, set the measurement matrix as Gaussian random measurement matrix with the size of 500×2500 and the sparse base matrix as Fourier sparse base matrix with the size of 2500×2500.

At the stage of hopping time estimation, set the original length of each observation object as 5000 points and the compression rate as 5, so the length after compression of each observation object is set to 1000 points. During the process of compressed sampling, set the measurement matrix as Gaussian random measurement matrix with the size of 1000×5000 and the sparse base matrix as an improved atomic dictionary with the size of 5000×5000.

To verify the effectiveness of hopping time estimation based on an improved atomic dictionary, select a section of observation object for simulation experiment under ideal channel conditions. The prior information is that the hopping point is the midpoint of the observation object. We search for all points in the observation object, and the matching condition of all points is shown in Figure 2.

It can be observed from Figure 2 that ${\left|\alpha_{1}\right|}^{2}+{\left|\alpha_{2}\right|}^{2}$ reaches its peak at the midpoint of the observation object; therefore, the estimated hopping time is the midpoint according to theorem 1, which is consistent with the actual result.

### 4.2. The Impact of AWGN on Proposed Algorithm

The simulation environment in this section is consistent with simulation 4.1, and AWGN is added to the ideal channel to study the impact of AWGN on proposed algorithm. To reduce search times, set a fixed search step size of 625 points. In order to measure the estimation effect, the estimation error analysis is conducted for the hopping time and center frequency of the two parameters. Equation (18) denotes the estimated mean-square error of hopping time, and Equation (19) denotes the estimated mean-square error of center frequency.
(18)$$\gamma_{t}=\frac{1}{N_{e}}{\sum_{n=1}^{N_{e}}\left(\frac{t_{h}-\hat{t_{h}}}{t_{h}}\right)}^{2}$$
(19)$$\gamma_{f}=\frac{1}{N_{e}}{\sum_{n=1}^{N_{e}}\left(\frac{f_{h}-\hat{f_{h}}}{f_{h}}\right)}^{2}$$
where $\gamma_{t}$ is the estimated mean-square error of hopping time, $t_{h}$ is the actual value of hopping time, $\hat{t_{h}}$ is the estimated value of hopping time, $\gamma_{f}$ is the estimated mean-square error of center frequency, $f_{h}$ is the actual value of center frequency, $\hat{f_{h}}$ is the estimated value of center frequency, and $N_{e}$ is the number of repeated simulations. The results obtained through 100 independent simulation experiments are represented in Figure 3a,b.

As shown in Figure 3a, the estimation accuracy of hopping time decreases as SNR decreases, because AWGN can affect the matching condition of all points in the observation object. Specifically, AWGN can introduce false peaks, and these false peaks interfere with the selection of the true peak, which is as shown in Figure 4. Moreover, as SNR decreases, the false peaks gradually approach and exceed the true peak, causing the false peaks to displace the true peak when selecting the peak.

As shown in Figure 3b, when SNR is less than −2 dB, the estimation accuracy of center frequency decreases as SNR decreases. This is because AWGN can affect the selection of the maximum dot product. Specifically, AWGN can increase the amplitude of the non-maximum dot product, which is as shown in Figure 5. Moreover, as SNR decreases, the amplitude of the non-maximum dot product gradually approaches and exceeds the amplitude of the maximum dot product, causing the non-maximum dot product to displace the maximum dot product when selecting the dot product.

### 4.3. The Impact of Fixed-Frequency Interference on the Proposed Algorithm

The simulation environment in this section is consistent with simulation 4.1, and fixed-frequency interference is added to the ideal channel to study the impact of fixed-frequency interference on the center frequency estimation performance of the proposed algorithm. The simulation is divided into two situations: one is that the frequency collision between the fixed-frequency interference and the FH signal occurs, and the other is that the frequency collision between the fixed-frequency interference and the FH signal does not occur. In both cases, the results obtained through 100 independent simulation experiments are shown in Figure 6.

As shown in Figure 6, when the signal-to-interference ratio (SIR) is greater than 14 dB, the center frequency estimation accuracy is the same in both cases. When SIR is less than 14 dB, the center frequency estimation accuracy in both cases decreases with the decrease in SIR, and the estimation accuracy of frequency collision is higher than that of frequency non-collision. The reason is that fixed-frequency interference can mask the FH signal, and the estimated frequency based on the maximum dot product is the frequency of fixed-frequency interference. At this time, if no frequency collision occurs, the estimated frequency will completely deviate from the actual frequency. If frequency collision occurs, the estimated frequency will partially deviate from the actual frequency, as shown in Figure 7.

It can be observed from Figure 7 that interference frequency components are not introduced in both high-SIR environments and low-SIR environments for frequency collision. For frequency non-collision, the introduced interference frequency components do not affect the extraction of center frequency components in a high-SIR environment, and the introduced interference frequency components will cover the center frequency components, affecting the extraction of center frequency components in a low-SIR environment.

### 4.4. The Impact of Search Step Size and Compression Ratio on the Proposed Algorithm

The simulation environment in this section is consistent with simulation 4.2, and the search step size is adjusted from 625 points to 1250 points and 250 points to study the impact of search step size on the estimation performance of the proposed algorithm for hopping time. The results obtained through 100 independent simulation experiments are shown in Figure 8.

As Figure 8 shows, when SNR is greater than 0 dB, the smaller the search step size, the higher the estimation accuracy of the hopping time, and when SNR is less than 0 dB, the smaller the search step size, the lower the estimation accuracy of the hopping time. This is because when SNR is greater than 0 dB, the smaller the search step size, the more the search points, the higher the time resolution, and the higher the estimation accuracy of the hopping time. However, the smaller the difference between ${\left|\alpha_{1}\right|}^{2}+{\left|\alpha_{2}\right|}^{2}$ corresponding to the two adjacent search data points, the more likely the noise will cause the false peak to cover the true peak. Therefore, when SNR is less than 0 dB, the smaller the search step size, the smaller the difference between ${\left|\alpha_{1}\right|}^{2}+{\left|\alpha_{2}\right|}^{2}$ corresponding to the two adjacent search data points, the worse the anti-noise performance, and the lower the estimation accuracy of the hopping time. The sensitivity of different search step size to noise is shown in Figure 9.

The sensitivity to noise means the deviation degree between the estimated value and actual value caused by noise. It can be observed from Figure 9 that the search step size of 1250 points has the lowest sensitivity to noise, while the enhancement of noise has the smallest impact on the matching situation of each point in the observation object, with the best anti-noise performance. The search step size of 250 points has the highest sensitivity to noise, while the enhancement of noise has the greatest impact on the matching situation of each point in the observation object, with the worst anti-noise performance.

The simulation environment in this section is consistent with simulation 4.2, and the compression ratio is adjusted from 5 to 2, 10 and 20 to study the impact of the compression ratio on the estimation performance of the proposed algorithm for center frequency. The results obtained through 100 independent simulation experiments are shown in Figure 10.

It can be observed from Figure 10 that as the compression ratio increases, the estimation accuracy of center frequency gradually decreases. This is because the higher the compression ratio, the smaller the amount of data obtained by compressed sampling, the less frequency information contained, and the worse the anti-noise performance, provided that the original data amount remains unchanged. Therefore, in a high-SNR environment, increasing the compression ratio appropriately can reduce the computational complexity of parameter estimation while maintaining high estimation accuracy, and in a low-SNR environment, decreasing the compression ratio appropriately can maintain high parameter estimation accuracy.

### 4.5. Comparison of Performance and Computational Complexity of Different Algorithms

This simulation will compare the estimation performance and computational complexity of the proposed algorithm with the algorithm in the literature [23] (compared algorithm). The compared algorithm and the proposed algorithm belong to the compressed sensing algorithm. For the estimation of hopping time, the proposed algorithm relies on improved atomic dictionary, and redundant atomic dictionary is the core of the compared algorithm. The simulation environment in this section is consistent with simulation 4.2, the search step size of the proposed algorithm is 625 points, and the compression ratio of these two algorithms is set to 8. The results obtained through 100 independent simulation experiments are shown in Figure 11a,b.

As shown in Figure 11a, in environments where SNR is greater than 0 dB, the estimation performance of the proposed algorithm for center frequency has significantly improved compared to the compared algorithm, with the estimated mean-square error reduced from the magnitude order of 10^−7^ to the magnitude order of 10^−9^. This is because the data length before compression for each segment of the FH signal in compared algorithm is 1024 points, while the data length before compression for each segment of the FH signal in the proposed algorithm is 2500 points. The larger the data length, the larger the frequency search range, and the higher the estimation accuracy of center frequency.

As shown in Figure 11b, the estimation performance of the proposed algorithm for hopping time has significantly improved compared to the compared algorithm. This is because the estimation of hopping time in the compared algorithm is based on redundant atomic dictionary, and the redundant atomic dictionary is based on the estimation of center frequency. In low-SNR environments, the estimation accuracy of center frequency is reduced, and the accuracy of the redundant atomic dictionary also decreases, which affects the estimation accuracy of hopping time. In contrast, the proposed algorithm is based on the improved atomic dictionary for the estimation of hopping time, which is not based on the estimation of center frequency, so it is not affected by the estimation accuracy of center frequency.

According to the computational complexity analysis in Section 3.4, the computational complexity comparison is shown in Table 2.

As shown in Table 2, in terms of the center frequency estimation, the computational complexity of the proposed algorithm is the same as that of the compared algorithm. In terms of hopping time estimation, the computational complexity of the proposed algorithm is slightly lower than that of the compared algorithm, with the difference $BN_{p}^{2}$ between the two.

## 5. Conclusions

In this paper, an FH signal parameter estimation algorithm based on the improved atomic dictionary is proposed. In this approach, the received FH signal is segmented and undergoes compressive sampling, and the center frequency is estimated by leveraging the position corresponding to the maximum dot product. To improve the temporal resolution of hopping time, the improved atomic dictionary is constructed to complete the temporal correction. The proposed algorithm makes full use of the property that frequency content at each time instant is intrinsically sparse, without reconstructing the FH signal and any prior information. It only needs the compressive measured values and improved atomic dictionary to estimate the parameters. Compared to the compressed sensing algorithm that combines a redundant atomic dictionary, the computational complexity is reduced, and the estimation accuracy of the parameters is significantly improved with the enhanced anti-noise performance. However, there is a lot of work to do to improve the anti-interference performance, especially in practical operations, which may face more complex interference. In addition, in the case of multiple FH signals, how to sort them and estimate their parameters is the next research direction.

## Figures and Tables

**Figure 1 sensors-23-05065-f001:**
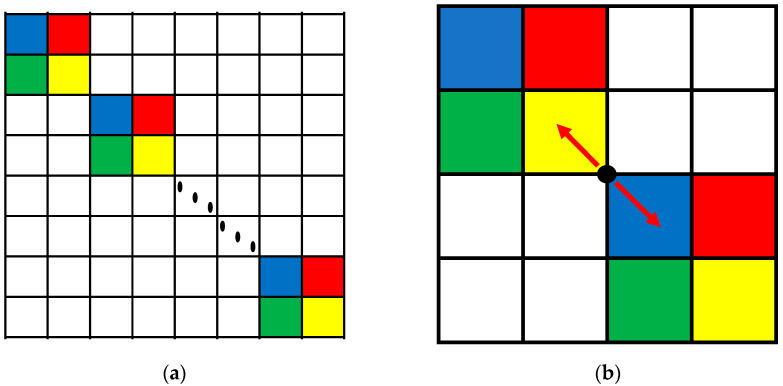
(**a**) The original improved atomic dictionary; (**b**) the improved atomic dictionary in the algorithm.

**Figure 2 sensors-23-05065-f002:**
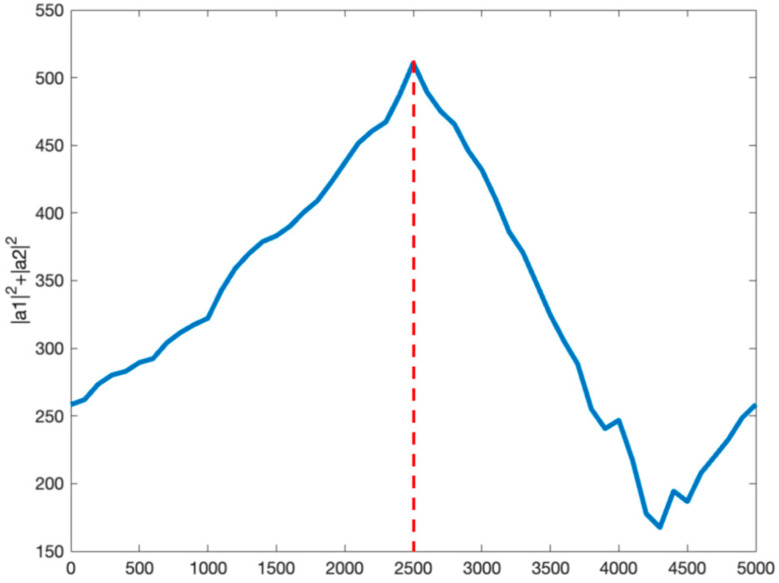
The matching condition of all points in the observation object.

**Figure 3 sensors-23-05065-f003:**
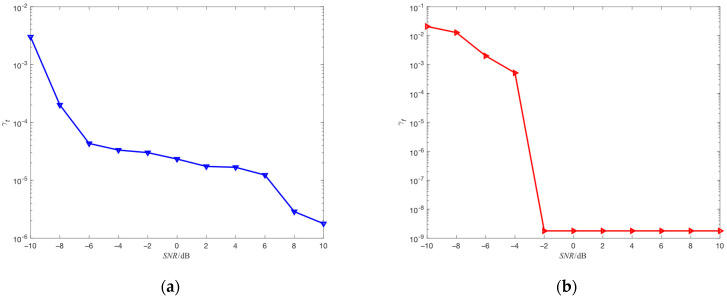
(**a**) The estimation performance of hopping time for different SNR; (**b**) the estimation performance of center frequency for different SNR.

**Figure 4 sensors-23-05065-f004:**
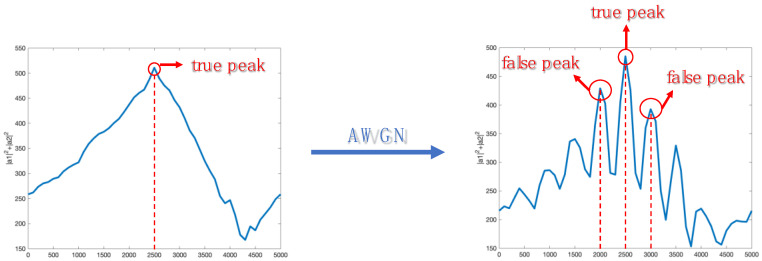
The matching condition of non-AWGN and AWGN.

**Figure 5 sensors-23-05065-f005:**
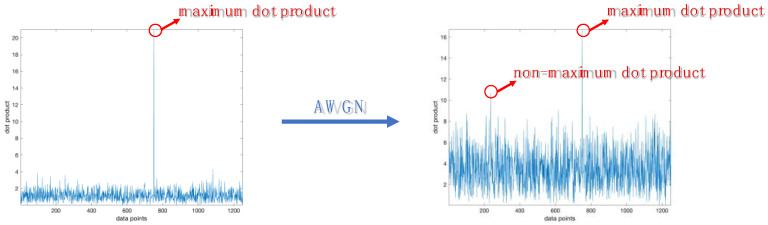
The dot product’s distribution of non-AWGN and AWGN.

**Figure 6 sensors-23-05065-f006:**
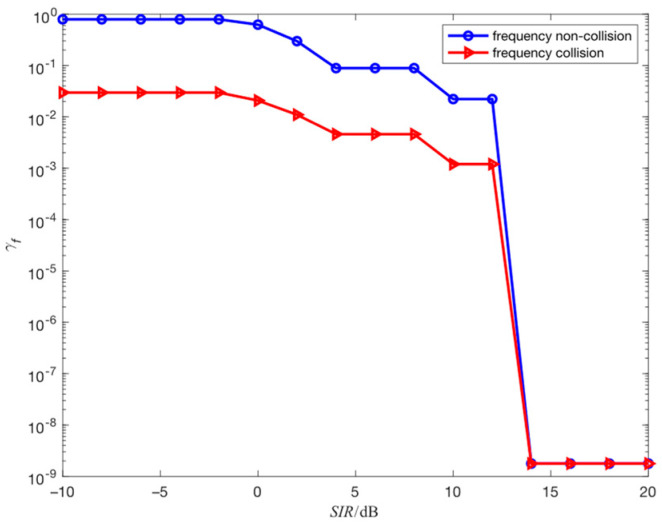
The estimation performance of center frequency for different SIR.

**Figure 7 sensors-23-05065-f007:**
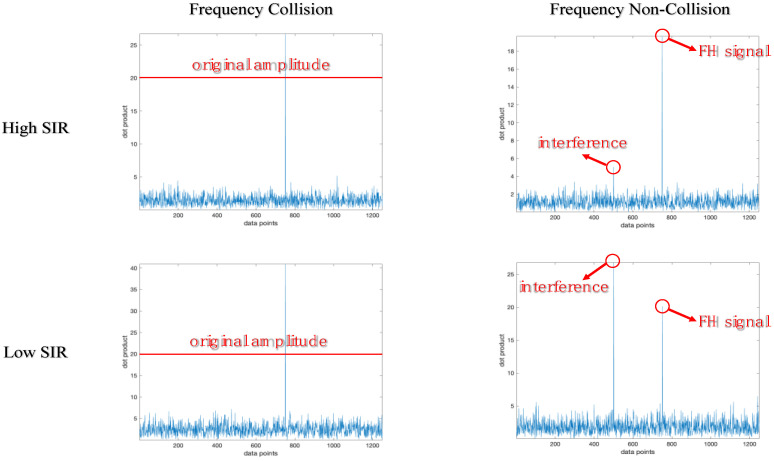
The distribution of dot product for different environments.

**Figure 8 sensors-23-05065-f008:**
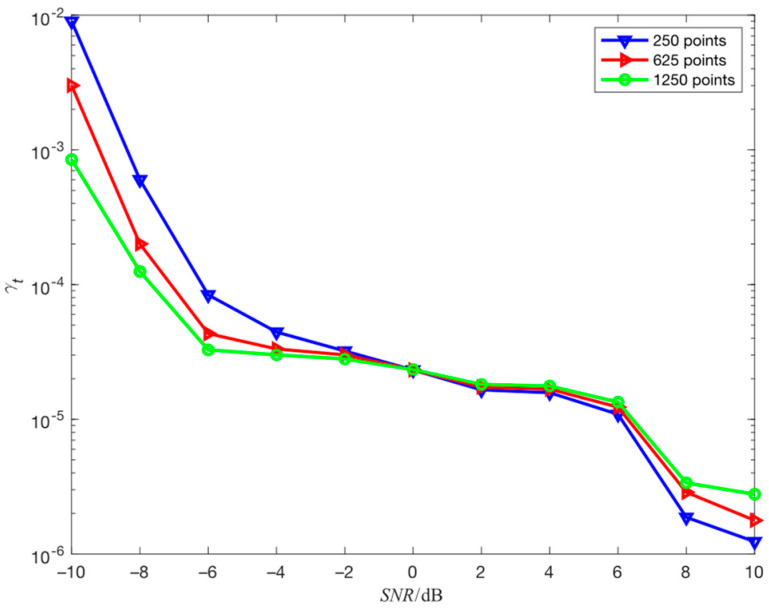
The influence of search step size on the estimation performance of hopping time.

**Figure 9 sensors-23-05065-f009:**
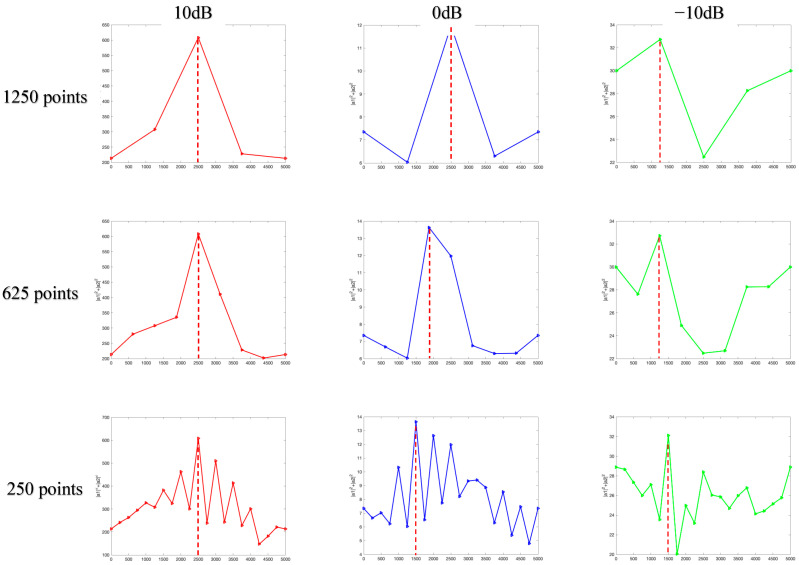
The matching condition for different SNR and search step size.

**Figure 10 sensors-23-05065-f010:**
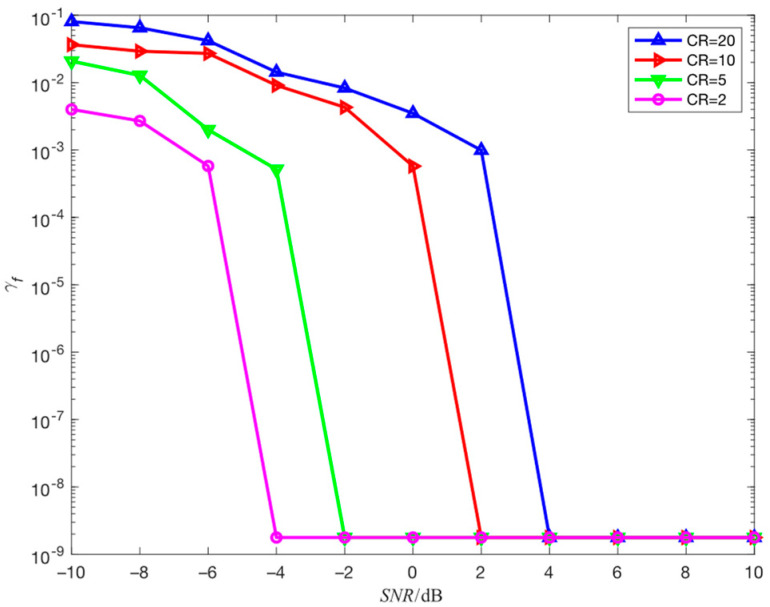
The influence of compression ratio on the estimation performance of center frequency.

**Figure 11 sensors-23-05065-f011:**
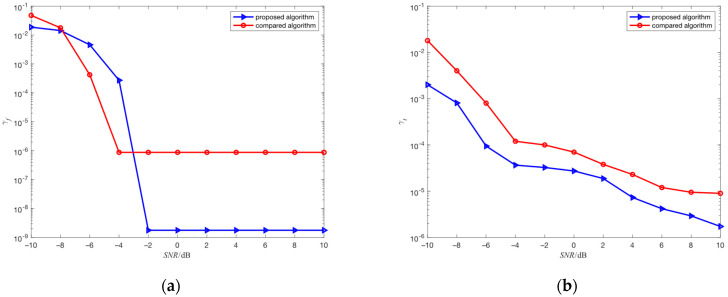
(**a**) The comparison of estimation performance of center frequency for different algorithms; (**b**) The comparison of estimation performance of hopping time for different algorithms.

**Table 1 sensors-23-05065-t001:** Simulation data setting.

Parameter	Value
Sampling Rate	1 MHz
Sampling Time	2 s
Symbol Rate	4 bit/s
Modulation Type	2 FSK
Carrier Frequency	10 Hz/20 Hz
Frequency Set	300 KHz~500 KHz
Pseudorandom Sequence	m sequence
Hop Rate	100 h/s
FH moment	5 ms

**Table 2 sensors-23-05065-t002:** The comparison of computational complexity for different algorithms.

	Representation	Computation $\left(k=8,\;M_{p}=7\right)$
Proposed Algorithm	$\frac{(L+8BM_{p})N_{p}^{2}}{k}$	$\frac{LN_{p}^{2}}{8}+7BN_{p}^{2}$
Compared Algorithm	$\frac{LN_{p}^{2}}{k}+8BN_{p}^{2}$	$\frac{LN_{p}^{2}}{8}+8BN_{p}^{2}$

## Data Availability

Not applicable.

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
