# Peer review of "Parameter Estimation Algorithm of Frequency-Hopping Signal in Compressed Domain Based on Improved Atomic Dictionary"

_sensors, 2023, doi:10.3390/s23115065_

Round 1

Reviewer 1 Report

This paper study the problem of estimating the parameters of frequency hopping signal using  improved atomic dictionary. Numerical results show that the proposed algorithm can achieve superior performance.

In my experience, the computation cost of atomic dictionary based method will be much larger than conventional ones. Thus, I recommend the authors to perform complexity analysis and show the rumtime cost the algorithms.

Reviewer 2 Report

This paper proposes an improved parameter estimation algorithm for FH signals. The experiment of the paper is sufficient and the structure is reasonable. I have the following suggestions:
1) How long does it take to ensure the accuracy in signal segment? Has the authors conducted any relevant experiments?
2) The image resolution is low, it is recommended to use vector images.
3) As non English speaking authors, it is recommended to use short sentences to increase readability.
4) Just from Figure 1 alone, there is no obvious difference in the improvement method and original method. It is recommended that the authors use a flowchart or provide examples to illustrate the proposed algorithm.
5) For the peak problem in Figures 4 and 5, it is recommended to try using meta-heuristic algorithms in the future.
6) From the experiment, I can not see the so-called "parameter" estimation in the title. What type of parameter did the authors estimate for the system? If it is an identification parameter, is there any statistical difference between its original value and the value after estimation.
7) For parameter estimation issues, more relevant papers can be consulted, such as https://doi.org/10.1007/s11071-021-06993-0 etc.

 Minor editing of English language required

Reviewer 3 Report

1. The detailed description of the compared algorithm and its design differences comparing with the proposed algorithm should be clarified;

2. the compression ratio CR used in the Fig.11 should be noted;

3. The parameter sensitivity analysis of the proposed estimation method should be added.

Minor editing of English language required. 
